# Generation of out-of-plane polarized spin current by spin swapping

Binoy K. Hazra [1,3], Banabir Pal [1,3], Jae-Chun Jeon[1], Robin R. Neumann [2], Börge Göbel [2], Bharat Grover[1], Hakan Deniz[1], Andriy Styervoyedov[1], Holger Meyerheim[1], Ingrid Mertig[2], See-Hun Yang[1] & Stuart S. P. Parkin [1] ✉

The generation of spin currents and their application to the manipulation of magnetic states is fundamental to spintronics. Of particular interest are chiral antiferromagnets that exhibit properties typical of ferromagnetic materials even though they have negligible magnetization. Here, we report the generation of a robust spin current with both in-plane and out-of-plane spin polarization in epitaxial thin films of the chiral antiferromagnet $Mn_3Sn$ in proximity to permalloy thin layers. By employing temperature-dependent spin-torque ferromagnetic resonance, we find that the chiral antiferromagnetic structure of $Mn_3Sn$ is responsible for an in-plane polarized spin current that is generated from the interior of the $Mn_3Sn$ layer and whose temperature dependence follows that of this layer's antiferromagnetic order. On the other hand, the out-of-plane polarized spin current is unrelated to the chiral antiferromagnetic structure and is instead the result of scattering from the $Mn_3Sn$/permalloy interface. We substantiate the later conclusion by performing studies with several other non-magnetic metals all of which are found to exhibit out-of-plane polarized spin currents arising from the spin swapping effect.

The Spin Hall effect (SHE)[1–4] allows for the generation of spin currents from charge currents that are passed through the interior of thin metallic layers. The magnitude and polarization direction of the spin current are widely inferred from the spin-torque ferromagnetic resonance technique (ST-FMR)[5–8] in which the spin current is used to provide torques on proximal ferromagnetic layers. The polarization of the spin current is typically observed to lie in the plane[9] of the metallic layer but the quest for out-of-plane polarized spin currents[10–18] has attracted much attention as they could be used to manipulate perpendicularly magnetized layers without any external magnetic field, an essential ingredient for spintronic applications.

The polarization direction of the spin current in conventional non-magnetic metals is always even under a magnetic field, but it has been predicted that currents passed through non-collinear antiferromagnet (AFM), such as the $Mn_3X$ (X = Sn, Ir), can give rise to additional spin currents that are odd under magnetic field[19,20]. Subsequently, a dominant odd magnetic SHE with an out-of-plane spin-polarization along

with a small even conventional SHE was inferred in single crystals of $Mn_3Sn$ from spin-pumping and ST-FMR experiments[21,22]. Although the magnetic SHE has been argued to be one of the fundamental mechanisms to generate out-of-plane polarized spin current in non-collinear antiferromagnetic thin films[23,24], its direct correlation with the antiferromagnetic order parameter by temperature dependent studies has not yet been established.

Here, we explore the SHE generated in high-quality epitaxial thin films of $Mn_3Sn$ (0001) via studies of the spin-orbit torque (SOT) on thin epitaxial layers of ferromagnet, permalloy ($Ni_{80}Fe_{20}$ = Py), which are grown on top of the $Mn_3Sn$ layers. We observe robust in-plane ($p_y$) and out-of-plane ($p_z$) polarized spin currents for various thicknesses of $Mn_3Sn$ when a charge current is passed through the $Mn_3Sn$/Py device (Fig. 1d). The in-plane (out-of-plane) polarized spin currents give rise to in-plane anti-damping-like (field-like) and out-of-plane field-like (anti-damping) torques, respectively. Both $p_y$ and $p_z$ retain the same magnitude and sign when large in-plane magnetic fields are applied to

[1]Max Planck Institute of Microstructure Physics, Weinberg 2, 06120 Halle (Saale), Germany. [2]Institut für Physik, Martin-Luther-Universität Halle-Wittenberg, 06099 Halle (Saale), Germany. [3]These authors contributed equally: Binoy K. Hazra, Banabir Pal. ✉e-mail: stuart.parkin@mpi-halle.mpg.de

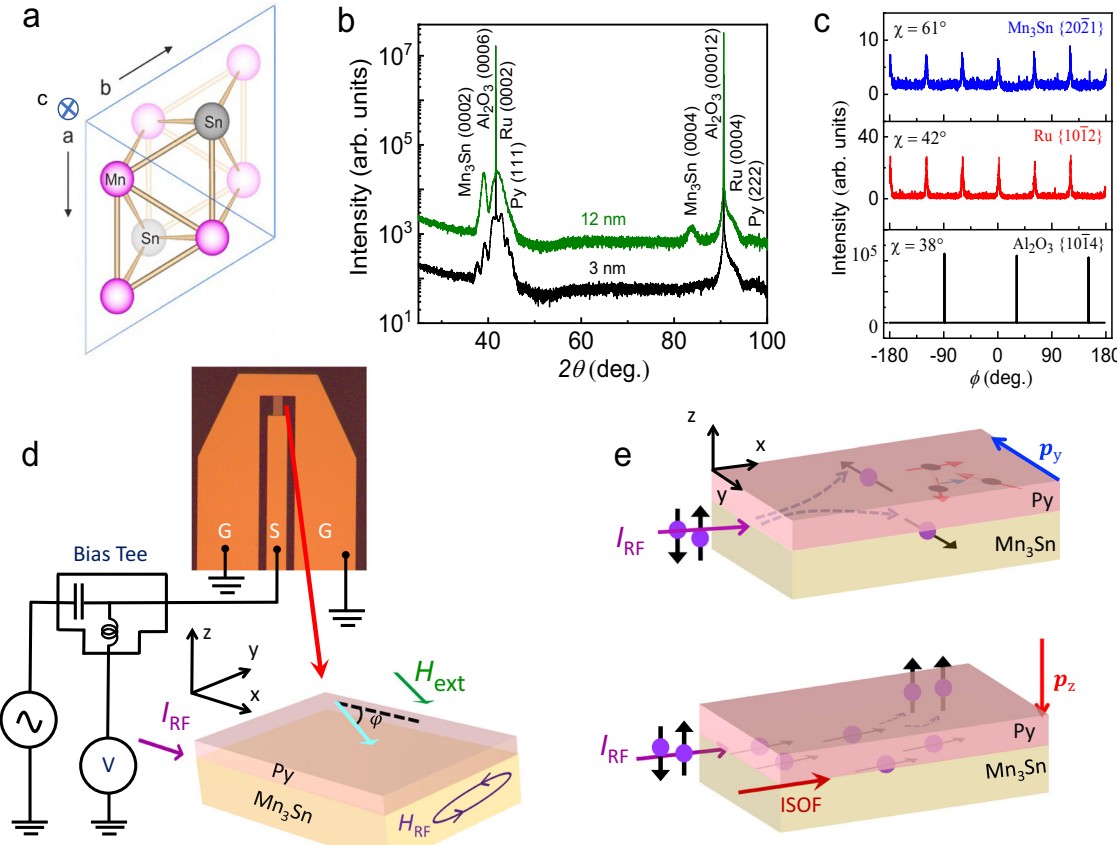

**Fig. 1 | X-ray diffraction, schematic of ST-FMR set-up and origin of spin polarizations. a** Structure of Mn₃Sn viewed along the [0001] direction. Pink and gray balls represent Mn and Sn atoms, respectively. **b** X-ray diffraction pattern of Ru/ Mn₃Sn (0001)/Py structures grown on Al₂O₃ (0001) substrates. Mn₃Sn layer thicknesses are 3 nm (black) and 12 nm (green). **c** Phi ($\phi$) scan across the Al₂O₃ {10$\bar{1}$4}, Ru {10$\bar{1}$2} and Mn₃Sn {20$\bar{2}$1} reflections for the Ru/Mn₃Sn (0001)/Py structures with 12 nm Mn₃Sn film. **d** Schematic of the ST-FMR experimental set-up

and Mn₃Sn/Py device. The directions of $I_{RF}$, $H_{ext}$ and angle ($\varphi$) between $I_{RF}$ and $H_{ext}$ are also shown. **e** The direction and origin of $p_y$ and $p_z$ for the Mn₃Sn/Py structure are shown schematically. The upper panel shows that $p_y$ originates from the SHE, which is related to the AFM structure of Mn₃Sn. $p_z$ arises when the spin-polarized current from Py is scattered at the interface due to the spin swapping effect (lower panel). Note that $\mathbf{p}_i = p_i\hat{\mathbf{i}}$ ($i = y$ or $z$). The interfacial spin-orbit field (ISOF) is also shown schematically.

reverse the AFM structure of the Mn₃Sn layer. By performing temperature-dependent ST-FMR, we demonstrate that $p_y$ strongly depends on the AFM structure whereas $p_z$ is unrelated to the antiferromagnetic ordering of Mn₃Sn, rather it originates from the scattering at the Py interface. We further substantiate the origin of $p_z$ by measurements on various non-magnetic metals (Cu, Ru, Re and Pt)/Py bilayers with different strengths of bulk spin–orbit coupling (SOC). Based on these experimental results, we demonstrate that this interface-scattered $p_z$ originates from the spin swapping mechanism[25–29].

## Results and discussion

Mn₃Sn crystallizes in a hexagonal $DO_{19}$ crystal structure with a kagome inverse triangular antiferromagnetic spin configuration within the (0001) plane (Fig. 1a) that is a result of the interplay between the geometric frustration of the spins coupled by Heisenberg exchange interaction and the Dzyaloshinskii-Moriya interaction. A small single-ion anisotropy results in a small in-plane moment[30], which allows for an external magnetic field to set the AFM structure in a specific domain structure. Epitaxial thin films of Mn₃Sn with thicknesses ($d_{AFM}$) ranging from 3 nm to 12 nm were grown on Ru buffered Al₂O₃ (0001) substrates using an ultra-high vacuum d.c. magnetron sputtering. A 5 nm thick Ni₈₀Fe₂₀ (Py) layer was sputtered on top of the Mn₃Sn layer and capped with a 3 nm thick TaN layer (see Methods). Figure 1b shows typical X-ray diffraction (XRD)

patterns of two representative samples with 3 nm and 12 nm thick Mn₃Sn layers. The (0002) and (0004) reflections from Ru and Mn₃Sn establish their epitaxial growth along (0001), which is further confirmed by transverse phi ($\phi$) scans of the Al₂O₃ {10$\bar{1}$4}, Ru {10$\bar{1}$2}, and Mn₃Sn {20$\bar{2}$1} reflections (Fig. 1c). A detailed structural analysis confirms that the crystal symmetry of Mn₃Sn belongs to P6₃/mmc (see section I, Supplementary Information (SI)). Atomic force micrographs display a smooth surface of all films with a typical roughness of -0.5 nm (Fig. S4a in SI). Transmission electron microscopy imaging confirms the high quality of the film structure (Fig. S4b in SI).

The properties of Ru/Mn₃Sn/TaN films grown without any Py layer were investigated by magnetization and transport measurements. A small magnetization within the kagome plane (0001) is observed when an in-plane magnetic field is used to orient the AFM into a single domain. The magnitude of this magnetization is consistent with the expected chiral AFM structure of the Mn₃Sn thin film[31] and, moreover, decreases as the temperature is increased close to the Néel temperature (Fig. S5a in SI). Again consistent with prior work[32] no anomalous Hall effect is observed when the current is applied within the kagome plane (Fig. S6b in SI).

The ST-FMR technique as shown in Fig. 1d was used to measure SOTs in GSG-type devices formed from the Ru(5 nm)/Mn₃Sn ($d_{AFM}$)/ Py(5 nm) structures (hereinafter referred to as Mn₃Sn/Py), which were fabricated to allow for the RF current ($I_{RF}$) to be oriented along two

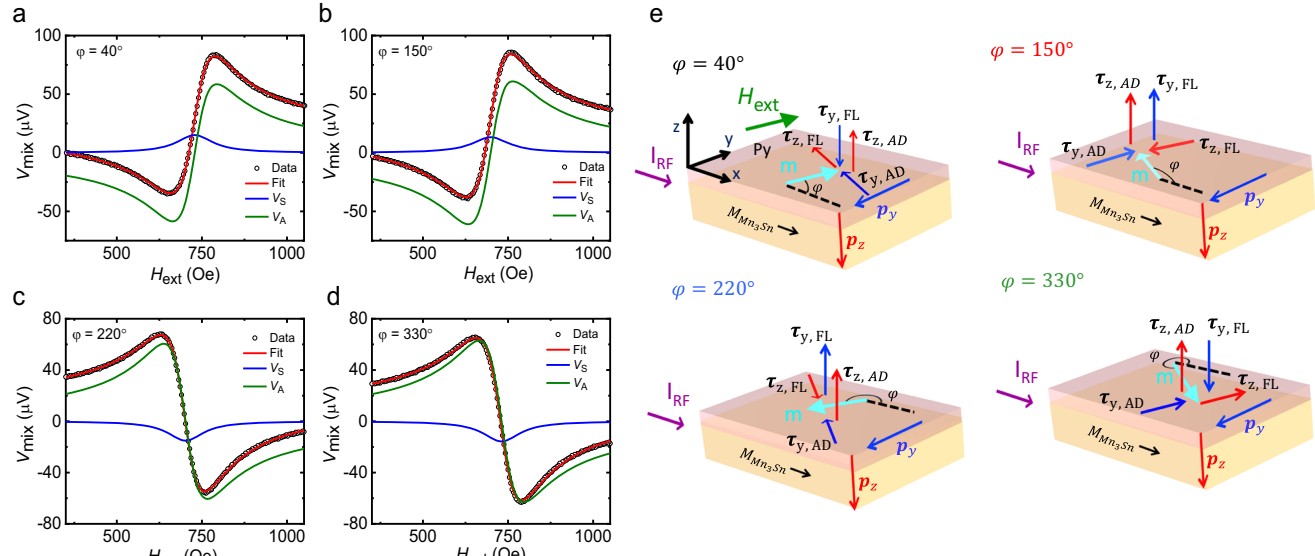

**Fig. 2 | ST-FMR d.c. voltage for different in-plane angles and torques directions.** **a–d** ST-FMR d.c. voltages, $V_{mix}$, along with the fits based on Eq. 1 are shown for the pristine Mn$_3$Sn(12 nm)/Py(5 nm) structures at $\varphi = 40°$, $150°$, $220°$ and $330°$, respectively. The individual $V_S$ and $V_A$ contributions are also plotted in the same figures. **e** Schematic illustration of $p_y$ and $p_z$ and respective torques due to these polarizations at the same $\varphi$. The vector forms of $\tau_{y,AD}$, $\tau_{y,FL}$ and $\tau_{z,AD}$, $\tau_{z,FL}$ are ($\mathbf{m} \times (\mathbf{m} \times \mathbf{p_y})$), ($\mathbf{m} \times \mathbf{p_y}$), ($\mathbf{m} \times (\mathbf{m} \times \mathbf{p_z})$) and ($\mathbf{m} \times \mathbf{p_z}$), respectively, where $\mathbf{m}$ is the magnetization of Py and $\mathbf{p_i} = p_i \hat{\boldsymbol{i}}$ ($i = y$ or $z$). Note that the in-plane moment of Mn$_3$Sn remains in the same direction for various $\varphi$ during ST-FMR measurements.

distinct in-plane crystallographic directions. The devices were prepared by standard optical lithography and a lift-off technique was used to prepare the Ti/Au electrical contacts (see Methods). $I_{RF}$ gives rise to a d.c. mixing voltage, $V_{mix}$, that is measured when a magnetic field is swept through the resonance condition for the Py. $V_{mix}$ is given by the equation[5,8],

$$V_{mix} = V_0 \left[ V_S \frac{\Delta H^2}{\Delta H^2 + (H_{ext} - H_{res})^2} + V_A \frac{\Delta H (H_{ext} - H_{res})}{\Delta H^2 + (H_{ext} - H_{res})^2} \right] \quad (1)$$

where $V_0$ is a constant pre-factor, $V_S$ and $V_A$ are the amplitudes of a symmetric and antisymmetric Lorentzian, respectively, $H_{res}$ is the resonance field, $\Delta H$ is the linewidth and $H_{ext}$ is the external magnetic field. First, we consider what we name the "0°" device in which $I_{RF}$ is along the in-plane crystallographic direction [01$\bar{1}$0] of Mn$_3$Sn. $V_{mix}$ is measured as a function of $\varphi$, the angle between $I_{RF}$ and $H_{ext}$. $V_S$ and $V_A$ are extracted by fitting $V_{mix}(\varphi)$ with Eq. (1). The magnitude of $V_S$ and $V_A$ are different at $\varphi = 40°$ and $150°$ compared to $\varphi = 220°$ and $330°$ (see Fig. 2a–d for $d_{AFM} = 12$ nm). Thus $V_S(\varphi)$ and $V_A(\varphi)$ display an asymmetric angular variation (Fig. 3a, b for $d_{AFM} = 12$ nm). Similar results are found for all $d_{AFM}$ (Fig. S8 in SI). By fitting $V_S(\varphi)$ and $V_A(\varphi)$ to the following equations[13], the in-plane and out-of-plane SOTs can be extracted:

$$V_S(\varphi) = -A(\tau_{x,AD} \sin\varphi \sin 2\varphi + \tau_{y,AD} \cos\varphi \sin 2\varphi + \tau_{z,FL} \sin 2\varphi) \quad (2)$$

$$V_A(\varphi) = -A\sqrt{1 + M_{eff}/H_{res}}(\tau_{x,FL} \sin\varphi \sin 2\varphi + \tau_{y,FL} \cos\varphi \sin 2\varphi + \tau_{z,AD} \sin 2\varphi) \quad (3)$$

where $A = -\frac{I_{RF}}{2} \frac{1}{\alpha(2\mu_0 H_{res} + \mu_0 M_{eff})}$ is a constant. $\tau_{i,AD}$ and $\tau_{i,FL}$ correspond to anti-damping-like and field-like torques resulting from the components of polarization, $p_i$ ($i = x,y,z$), of the spin current along z. We find that $\tau_{x,AD}$ and $\tau_{x,FL}$ due to $p_x$ are negligibly small whereas the magnitude of $\tau_{y,AD}/\tau_{y,FL}$ is large compared to $\tau_{z,AD}/\tau_{y,FL}$ and $\tau_{z,FL}/\tau_{y,FL}$. Note that $\tau_{y,AD}$, $\tau_{z,AD}$ and $\tau_{z,FL}$ are normalized to $\tau_{y,FL}$, which is dominated by Oersted field, for ease of comparison with

other systems that we discuss later. Hereinafter, the absolute value of $\tau_{y,AD}/\tau_{y,FL}$, $\tau_{z,AD}/\tau_{y,FL}$ and $\tau_{z,FL}/\tau_{y,FL}$ are referred to as $\tau'_{y,AD}$, $\tau'_{z,AD}$ and $\tau'_{z,FL}$, respectively. Note that the effective magnetization which is used to calculate the torques, is independent of $d_{AFM}$ (Fig. S7b in SI). We also note that the addition of a spin-pumping contribution[33] to $\tau_{y,AD}$ does not reproduce the unusual variation of $V_S(\varphi)$ (Fig. S8 in SI). The directions of $\tau_{y,AD}$, $\tau_{y,FL}$, $\tau_{z,AD}$ and $\tau_{z,FL}$ are shown schematically in Fig. 2e for several $\varphi$. The interplay between $\tau_{y,AD}$ and $\tau_{z,FL}$ ($\tau_{y,FL}$ and $\tau_{z,AD}$) at different $\varphi$ results in the unusual variation of $V_S(\varphi)$ ($V_A(\varphi)$). In the second set of "90°" devices $I_{RF}$ is along the in-plane crystallographic direction [2$\bar{1}\bar{1}$0] of Mn$_3$Sn (Fig. 3c, d). Note that the angle between the [2$\bar{1}\bar{1}$0] and [01$\bar{1}$0] directions is 90°. $\tau'_{y,AD}$ remains unchanged whereas $\tau'_{z,AD}$, $\tau'_{z,FL}$ are reduced to a small but non-zero value (Fig. S10, in SI). Previously, it has been shown in other materials that $p_z$ displays a cosine angular dependence as a function of $\varphi$ when $p_z$ originates due to either low crystal symmetry or low magnetic symmetry[10,14,15]. Theoretically, the same cosine angular dependence of $p_z$ is also calculated for Mn$_3$Sn assuming a chiral AFM structure (see section VI in SI). Thus, the finite $\tau'_{z,AD}$ and $\tau'_{z,FL}$ that we measure for the 90° device necessitates a distinct origin for $p_z$.

We find that neither $\tau'_{y,AD}$ nor $\tau'_{z,AD}$ and $\tau'_{z,FL}$ have a significant dependence on $d_{AFM}$, which suggests that they arise from the interface between Mn$_3$Sn and Py (Fig. 4a–c) or, equivalently, that the spin diffusion length in Mn$_3$Sn is small[34]. The 6-fold symmetry of the AFM structure of Mn$_3$Sn means that there are six equivalent magnetic ground states[35]. The AFM structure of Mn$_3$Sn can be set in a single domain state (or its twin) by applying a sufficiently large in-plane positive (or negative) magnetic field[36]. Here using $H_{set} = \pm 7$ T at 300 K we find, however, that $V_S(\varphi)$ and $V_A(\varphi)$ remain the same (Fig. S9 in SI). Consequently, the strength of $\tau'_{y,AD}$, $\tau'_{z,AD}$ and $\tau'_{z,FL}$ are unchanged for the twin AFM domains, which shows that both $p_y$ and $p_z$ are even under magnetic field. Since thickness and external magnetic field dependence of SOTs do not provide an insight into the mechanism of $p_y$ and $p_z$, we explore the temperature dependence of the SOTs close to the Néel temperature of Mn$_3$Sn ($T_N = 420$ K). Interestingly, we find that

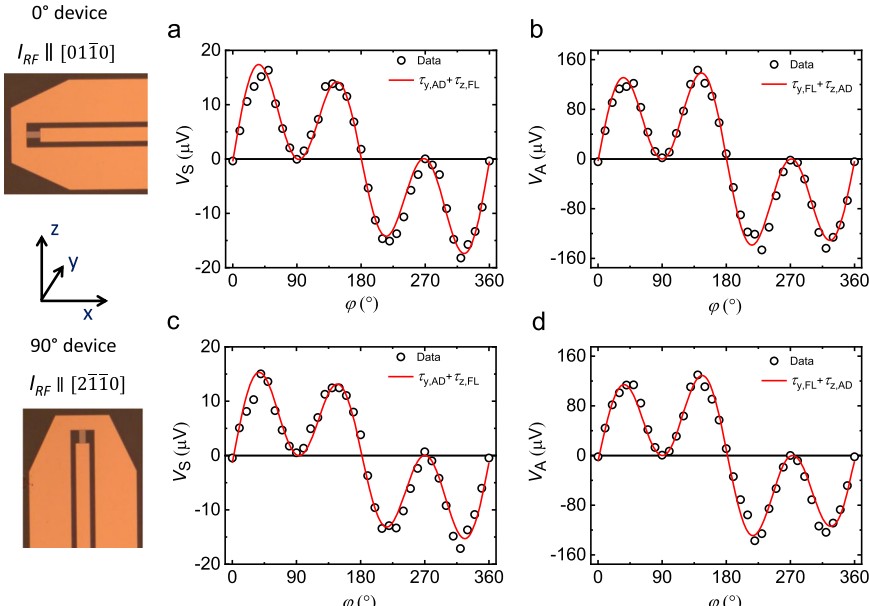

**Fig. 3 | Angular variation of $V_S$ and $V_A$ for two different devices. a, b** Angular variation of $V_S$ and $V_A$ along with the fits based on Eqs. (2, 3) are shown in the range of 0° to 360° for the Mn₃Sn(12 nm)/Py(5 nm) film. Here $I_{RF}$ is along the in-plane crystallographic direction [01$\bar{1}$0] of Mn₃Sn. **c, d** Angular variation of $V_S$ and $V_A$ along with the fits are shown for the 90° device where $I_{RF}$ is along the in-plane crystallographic direction [2$\bar{1}\bar{1}$0] of Mn₃Sn. The optical images of both the devices are also shown.

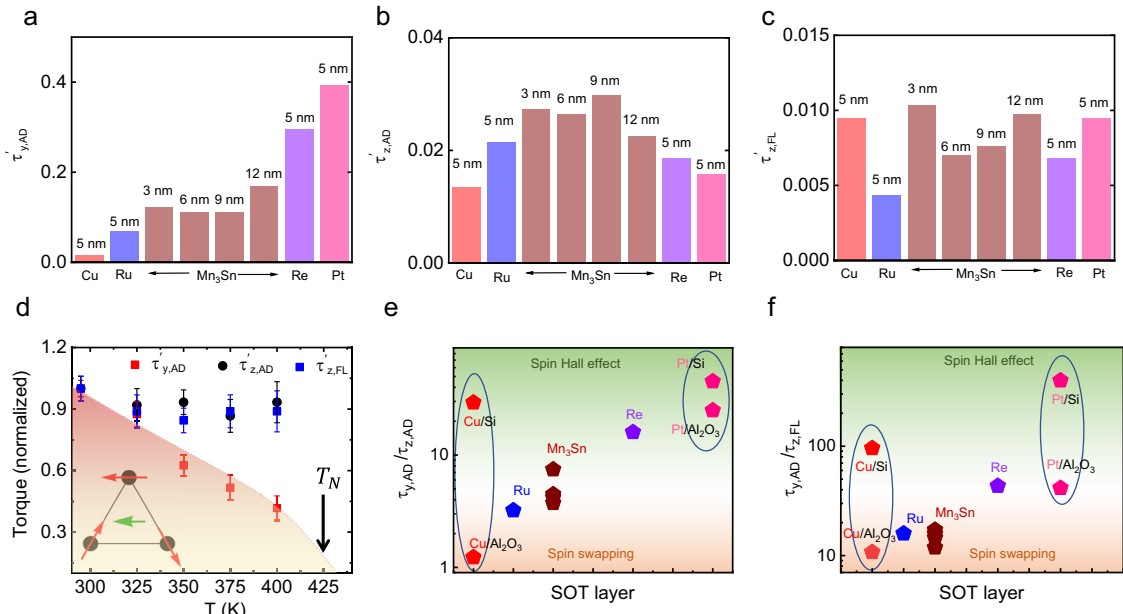

**Fig. 4 | Normalized torques, their temperature dependence and determination of SHE and spin swapping contribution. a–c** $\tau'_{y,AD}$ ($|\tau_{y,AD}/\tau_{y,FL}|$), $\tau'_{z,AD}$ ($|\tau_{z,AD}/\tau_{y,FL}|$) and $\tau'_{z,FL}$ ($|\tau_{z,FL}/\tau_{y,FL}|$) for Cu(5 nm)/Py(5 nm), Ru(5 nm)/Py(5 nm), Mn₃Sn ($d_{AFM}$ = 3-12 nm)/Py(5 nm), Re(5 nm)/Py(5 nm) and Pt(5 nm)/Py(5 nm) films, which are grown on Al₂O₃ substrate. **d** Temperature dependence of $\tau'_{y,AD}$, $\tau'_{z,AD}$ and $\tau'_{z,FL}$ for the Mn₃Sn(12 nm)/Py(5 nm) film. **e, f** The absolute value of $\tau_{y,AD}/\tau_{z,AD}$ and $\tau_{y,AD}/\tau_{z,FL}$ for Mn₃Sn/Py and non-magnetic (Cu, Ru, Re and Pt)/Py bilayers. SHE and spin-swapping contribute to all the bilayers and SHE dominates in Pt and Re whereas spin-swapping influences in Cu and Ru. As a function of disorder, spin swapping is large for Cu/Py and Pt/Py bilayers grown on Al₂O₃ substrates whereas SHE is relatively significant when the same structures are grown on Si/SiO₂ substrates. Note that the value of $\tau_{y,AD}/\tau_{z,AD}$ and $\tau_{y,AD}/\tau_{z,FL}$ for Al₂O₃/Cu/Py structure is multiplied by 10 for better visibility compared to other data points. All the data presented in this figure are measured on 0° device.

$\tau'_{y,AD}$ is significantly reduced with increase in temperature (Fig. 4d). This is a clear evidence that $\tau'_{y,AD}$ is directly related to the AFM structure of Mn₃Sn. With increase in temperature, AFM domains start to fluctuate which causes a reduction in $\tau'_{y,AD}$. Although $p_y$ arises from the AFM structure, it shows a thickness independent behavior due to the small spin diffusion length of Mn₃Sn[34]. On the other hand, $\tau'_{z,AD}$ and

$\tau'_{z,FL}$ remain unchanged in the temperature range of 300-400 K (Fig. 4d), which implies that $\tau'_{z,AD}$ and $\tau'_{z,FL}$ are unrelated to the AFM structure of Mn₃Sn. The temperature and thickness independent behavior of $\tau'_{z,AD}$ and $\tau'_{z,FL}$ indicates the interfacial origin of $p_z$, which is distinct compared to recent reports on the non-collinear AFM Mn₃GaN[13].

To investigate how the Mn$_3$Sn/Py interface influences $p_z$, we modified the interface by inserting a 2 nm thick Cu layer. The magnitude of $V_S(\varphi)$ and the strength of $\tau'_{y,AD}$, $\tau'_{z,AD}$ and $\tau'_{z,FL}$ remain similar to the structure without the Cu insertion layer (Fig. S14a, b in SI). This is because $\tau_{y,AD}$ is negligibly small in Cu. On the other hand, $\tau_{y,AD}$ is large in Pt and indeed when a Pt layer (2 nm thick) is inserted the magnitude of $V_S(\varphi)$ is significantly modified (Fig. S14c, d in SI). We also find that $\tau'_{y,AD}$ is enhanced due to the additional $\tau_{y,AD}$ from Pt, but $\tau'_{z,AD}$ and $\tau'_{z,FL}$ remain almost unchanged. The fact that $\tau'_{z,AD}$ and $\tau'_{z,FL}$ are nearly equal for both 2 nm Cu and Pt insertion layers shows that the Py layer plays a key role in generating $p_z$.

There are two distinct theories that account for the origin of $p_z$ related to a proximal in-plane magnetized layer. Amin et al.[37,38] predict that an in-plane magnetized ferromagnetic (FM) layer proximal to a non-magnetic layer (NM) can generate a $p_z$ as observed experimentally in FM$_1$/NM/FM$_2$ tri-layers[39]. The bottom FM$_1$ layer generates an in-plane spin-polarized current which is transmitted through the NM layer and gives rise to $p_z$ that applies torques on the top FM$_2$ layer. However, in our case, we only have a single FM layer. Thus, the second model in which the generation of $p_z$ can be understood from a spin swapping mechanism[25–29] in a SOT layer/FM bilayer structure is more relevant. The spin-polarized currents (along 'x') from the proximal FM are reflected from the interface (between SOT layer and FM) and, simultaneously, precess in the presence of an interfacial spin-orbit field (lower panel in Fig. 1e). After the precession, the primary spin-polarized current generates a secondary spin current with out-of-plane spin-polarization, $p_z$ which successively induces substantial torques.

The spin-swapping effect is differentiated by the presence of a $\tau_{z,AD}$-like torque and $\tau_{z,FL}$-like torque as opposed to the $\tau_{y,AD}$-like torque due to the SHE. The ratios $\tau_{y,AD}/\tau_{z,AD}$ and $\tau_{y,AD}/\tau_{z,FL}$ thus indicate the dominant mechanism whether arising from spin swapping or the SHE. To corroborate this, we have replaced Mn$_3$Sn with various non-magnetic metals with a wide range of SOC strengths, which includes Cu, Ru, Re, and Pt with the expectation that as the SOC increases the nature of the torque will show a crossover from a pure spin swapping regime to an SHE dominating regime. In our experiments, indeed we find that $\tau_{y,AD}/\tau_{z,AD}$ and $\tau_{y,AD}/\tau_{z,FL}$ are minimal for small SOC metals such as Cu and Ru (Fig. 4e, f). On the other hand, heavy metals such as Re and Pt display a significant $\tau_{y,AD}/\tau_{z,AD}$ and $\tau_{y,AD}/\tau_{z,FL}$ due to the dominance of an SHE. Interestingly, we find that Mn$_3$Sn lies in the transition region between these two mechanisms (Fig. 4e, f). Even though $\tau_{y,AD}/\tau_{z,AD}$ and $\tau_{y,AD}/\tau_{z,FL}$ increase monotonically with SOC, the magnitude of $\tau'_{z,AD}$ and $\tau'_{z,FL}$ is almost independent for different transition metals (Fig. 4b, c), indicating that the spin-swapping effect is nearly constant for the elements considered. This independent behavior of $\tau'_{z,AD}$ and $\tau'_{z,FL}$ further shows that the generation of $p_z$ is controlled by the proximal Py layer. On the other hand, the SHE causes an enhancement of $\tau'_{y,AD}$ as a function of their SOC (Fig. 4a). Furthermore, theoretical predictions also show that the creation of $p_z$ via a spin-swapping process is more effective when disorder at the interface is reduced. Indeed, we find that $\tau_{y,AD}/\tau_{z,AD}$ and $\tau_{y,AD}/\tau_{z,FL}$ are significantly larger when Cu/Py and Pt/Py are grown on Si/SiO$_2$ with a lower degree of crystallinity whereas $\tau_{y,AD}/\tau_{z,AD}$ and $\tau_{y,AD}/\tau_{z,FL}$ are smaller when highly crystalline Cu/Py and Pt/Py are grown on an (0001) oriented Al$_2$O$_3$ single crystalline substrate (Fig. 4e, f). Additionally, we have replaced the Py layer in Al$_2$O$_3$/Cu/Py structures with Fe and Ni to tune the magnetization of the FM in order to investigate the dependence of the SOTs on the strength of the magnetization. Interestingly, we do see a relatively weak magnetization dependence of $\tau'_{z,AD}$ and $\tau'_{z,FL}$ whereas $\tau'_{y,AD}$ strongly depends on the magnetization (see Fig. S18 in SI). Recently, the experimental and theoretical development of the orbital Hall effect (OHE) shows the existence of only $p_y$ polarization, which strongly depends on the magnetization of the FM[40–43]. Assuming the same analogy if the OHE generates $p_z$, it should strongly depend on the magnetization of the ferromagnets. The weak magnetization dependence of $\tau'_{z,AD}$ and $\tau'_{z,FL}$ rules out $p_z$ being derived from an OHE and rather further supports the spin-swapping mechanism. Note that it has been predicted theoretically that $p_z$ generated by spin swapping shows a weak magnetization dependence[28], which we observe in our study.

## Conclusion

In summary, we have shown the presence of robust and thickness independent $p_y$ and $p_z$ in Mn$_3$Sn/Py bilayers. From their temperature dependence, we have demonstrated that $p_y$ originates from the AFM structure of Mn$_3$Sn whereas $p_z$ is unrelated to the AFM structure and is generated at the Py interface. Moreover, the observation of $p_z$ when Mn$_3$Sn is replaced by several different non-magnetic metals, i.e., Cu, Ru, Re, and Pt is consistent with an interfacial origin by a spin swapping mechanism. This work provides insights into the origin of unconventional spin polarizations not only in chiral non-collinear antiferromagnets but also in various non-magnetic metals. Our observation of interface-scattered $p_z$ will further enrich the field of spin-orbitronics.

## Methods

### Sample preparation

All the thin-films were grown by d.c. magnetron sputtering in an ultra-high vacuum system with a base pressure of $1 \times 10^{-9}$ Torr. Atomically flat Al$_2$O$_3$ (0001) substrates were prepared by a wet etching procedure followed by a heat treatment at 1200 °C for 4 h. A 5 nm thick Ru buffer layer was first sputtered onto an Al$_2$O$_3$ (0001) substrate at ambient temperature using a sputtering power of 15 W and an Ar pressure of 3 mTorr. The Mn$_3$Sn layer was formed by co-sputtering of Mn and Sn from elemental sputter targets onto the Ru (0002) buffer layer at 200 °C and at an Ar pressure of 3 mTorr. An optimized composition of Mn$_{3.1}$Sn was used where the excess Mn helps to stabilize the hexagonal $D0_{19}$ phase. The sputtering powers were of 43 W and 8 W, respectively, for Mn and Sn. The Mn$_3$Sn layer thickness was varied from 3 to 12 nm. Cu, Ru, Pt and Re films, with layer thicknesses of ~5 nm were grown at room temperature directly onto Al$_2$O$_3$ (0001) substrates using sputtering powers of 30 W, 15 W, 30 W and 30 W, respectively, and at an Ar pressure of 3 mTorr. A 5 nm thick Ni$_{80}$Fe$_{20}$ (Py) layer was sputtered on top of Mn$_3$Sn and other non-magnetic metals with a sputtering power of 30 W and an Ar pressure of 3 mTorr. Also, Cu(5 nm)/Py(5 nm) and Pt(5 nm)/Py(5 nm) were sputtered on Si (001)/SiO$_2$ (25 nm) at room temperature. All the films were capped with a 3 nm thick TaN layer to prevent oxidation.

### Device fabrication

The films were patterned into ST-FMR devices (Fig. 1d) oriented along different crystallographic directions with device lengths (75 μm) and widths (25 μm) using conventional photolithography techniques (365 nm maskless laser writer; MLA150, Heidelberg). Etching was carried out using Ar ion beam milling and in-situ secondary ion mass spectroscopy was used for end point detection. Electrical contacts were formed using magnetron sputtered Ti (2 nm)/Au (100 nm) bilayers.

### Sample characterization

Rutherford backscattering (RBS) was used to determine the composition of Mn$_3$Sn film. The crystal structure of the films was characterized using a Bruker D8 Discover diffractometer with Cu K$_\alpha$ source and Gallium-Jet X-ray source. Atomic force microscopy measurements were performed to probe the surface topography. A FEI TITAN 80–300 transmission electron microscope (TEM) with a probe corrector was used at an accelerating voltage of 300 kV for scanning TEM studies. The magnetic properties of films were characterized using a Quantum Design MPMS3 SQUID magnetometer. Temperature-dependent measurements of the longitudinal resistivity and the anomalous Hall resistivity were carried out in a Quantum Design Physical Property Measurement System (PPMS). Spin-orbit torques were measured using

a home-made spin-torque ferromagnetic resonance set-up at 300 K and another set-up in a Lake-Shore probe station for temperature dependence measurements in the range of 300 K-400 K.

### ST-FMR measurements

In the ST-FMR measurements, an RF current ($I_{RF}$) with power of 21 dBm at different frequencies was applied and the in-plane magnetic field was swept between 1200 Oe to 0. $V_{mix}$ at $\varphi = 45°$ was measured at different frequencies to calculate $M_{eff}$ and Gilbert damping constant. Using $M_{eff}$ and $H_{res}$ for a particular frequency ($f = 8$ GHz), the strengths of individual torques are calculated from the fit of $V_S(\varphi)$ and $V_A(\varphi)$ using Eqs. (2, 3). Note that we did not evaluate the constant 'A' as all the torques are finally normalized by $\tau_{y,FL}$ for ease comparison. For the temperature dependence, the temperature was stabilized at a particular temperature then the angular dependence was carried out to evaluate the torques.

### Data availability

The Source Data underlying the figures of this study are available online. Source data are provided with this paper.

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

## Acknowledgements
We thank Ilya Kostanovskiy for the help with RBS analysis. S.S.P.P. acknowledges funding from the European Union's Horizon 2020 research and innovation program under grant agreement no. 766566 (ASPIN), the Alexander von Humboldt Foundation in the framework of the Alexander von Humboldt Professorship endowed by the Federal Ministry of Education and Research, and the Deutsche Forschungsgemeinschaft (DFG, German Research Foundation)—project no. 403505322, Priority Program (SPP) 2137. I.M. acknowledges support from the DFG under SFB TRR 227.

## Author contributions
S.S.S.P. conceived the project. B.K.H. prepared the samples with the help of A.S. B.P. and J.C.J. fabricated the devices. B.K.H. and B.P. carried out all the experiments with help of J.C.J. and B. Grover, B.K.H., B.P., and S.H.Y. analyzed the data. H.D. performed TEM experiments. H.M. carried out crystallographic symmetry analysis. R.R.N., B. Göbel, and I.M. performed theoretical calculations. All authors discussed the results. B.K.H., B.P., and S.S.S.P. wrote the manuscript with substantial contributions from all authors.

## Funding

## Competing interests
The authors declare no competing interests.
