## [Peer Review File · Nature Communications]

Reviewers' Comments:

Reviewer #1:

In this work, B. K. Hazra et al reported interface-induced out-of-plane spin polarization (p_z) in noncollinear antiferromagnet Mn_3Sn and nonmagnets Cu, Ru, Re and Pt. By performing temperature dependent spin-torque ferromagnetic resonance (ST-FMR) measurements, this work demonstrated that the interfaces of Mn_3Sn/Py or NM/Py is the origin of p_z , and the generation interface can be ascribed to the spin swapping effect. The generation of p_z is highly desired and the physical mechanism is still not cleared up to now. This work is timely because it attempts to investigate the interfacial origin of p_z , and the results are very interesting. However, several questions need to be addressed before I formally recommend this work to be published in Nature Communications.

1. As shown in Figure 3, p_z exists at both 0° and 90° devices. Whether p_z also exists in a device with specific angle (such as 45°)? What's the relationship between p_y , p_z and the crystal orientation (i.e. device angle)?
2. Because p_z is different for the 0° and 90° devices (as shown in Fig. 3), whether the result in Fig.4 is from 0° or 90° device should be described in the figure caption.
3. In this work, the ST-FMR device is the GS-type instead of the standard GSG-type (as shown in Ref. 10). Note that GS-type device may produce misleading signals or even artifacts [JMMM **505** 166727 (2020)]. I strongly recommend authors to use the standard GSG-type devices to test the ST-FMR results.
4. The discussion about the generation mechanism of p_z is not vivid enough. Thus, I recommend authors to add a schematic of the spin swapping mechanism at the interface of Mn_3Sn and Py , which is helpful for readers to follow.
5. This work uses $\tau_{y,AD}/\tau_{y,FL}$ and $\tau_{z,FL}/\tau_{y,FL}$ to represent the intensity of p_y and p_z . While I recommend authors to calculate the values of spin torque efficiency of p_y and p_z , as done in Ref. 15.
6. According to the temperature dependent experiments, the generation of p_y in Mn_3Sn is mainly caused by the SOC-independent AFM structure, rather than conventional SOC-dependent SHE. Thus, I think symbols of Mn_3Sn is not suitable in Fig. 4d considering the x-label is SOC.

Reviewer #2:

Remarks to the Author:

The manuscript is devoted to spin currents in epitaxial thin films of the chiral antiferromagnet Mn₃Sn interfaced with a thin layer of permalloy. By performing spin-torque ferromagnetic resonance (FMR) measurements, the authors find two types of spin currents – polarized in-plane and polarized out-of-plane. Based on the temperature-dependent FMR results for Mn₃Sn/permalloy bilayers and the results of studies with several other non-magnetic metals at the interface, the authors conclude that the in-plane polarized spin current is generated from the interior of the Mn₃Sn layer, while the out-of-plane polarized spin current originates from scattering at the Mn₃Sn/permalloy interface. The authors put forward the spin swapping effect as the origin of the observed out-of-plane polarized spin current.

Generation and understanding the nature of spin polarized spin currents in magnetic thin-film systems is important due to these currents being able to produce spin torques and switch magnetization of a ferromagnetic layer. Several recent publications have emphasized the importance of the out-of-plane polarized spin current, due to its efficiency in switching perpendicular magnetized films in the absence of an external magnetic field. In case of chiral antiferromagnets, such as Mn₃Sn, the origin of the out-of-plane polarized spin current has been attributed to the non-collinear antiferromagnetic order of these antiferromagnets. The authors of the manuscript challenge this interpretation, by showing that the out-of-plane polarized spin current is correlated not with the antiferromagnetic order in Mn₃Sn but rather with the strength of spin-orbit coupling at the interface. This finding is important for elucidating the origin of the out-of-plane polarized spin current capable of deterministic switching a perpendicular magnetization. I would recommend publication of this manuscript after the authors have addressed the two following issues.

1. The authors explain their observation in terms of the spin swapping effect and even include this term in the title of the manuscript. While this explanation may be plausible, in my view, there is not enough evidence provided in the manuscript to make this interpretation solid. Other possible origins of the out-of-plane polarized spin current not related to the antiferromagnetic order may play a role, such as an interface-generated spin current, a spin-orbit self-torque due to a composition gradient in the ferromagnetic layer (e.g., due to intermixing at the interface), an orbital Hall effect, etc. I think that the authors need to relax their statements and use the spin swapping effect as a possible explanation rather than the proved finding. I also think that "spin swapping" should be removed from the title of the manuscript.

2. Previous experimental results have shown that the spin-orbit torque generated in a Mn₃Sn/ferromagnet bilayer can be reversed with the reversal of the antiferromagnetic order, i.e. it is odd with respect to time reversal symmetry. The authors are saying however that their torques are even under magnetic field. Why the results presented in the manuscript are different from the previous observations?

Reviewer #3:

Remarks to the Author:

B. K. Hazra et al., reported the experimental study of out-of-plane polarized spin current by spin swapping. The spin torque due to in-plane and out-of-plane spin polarization was measured utilizing the ST-FMR method in Py/Epitaxial thin Mn₃Sn film bilayer. Temperature dependence above Neel temperature suggests the in-plane spin polarization is related to the chiral spin structure, however, the out-of-plane polarization is not related to the spin structure. By control experiments using several nonmagnetic metals, they conclude that the out-of-plane components are caused by the spin-swapping effect in Mn₃Sn.

Because the noncollinear antiferromagnet could enable new functionality in spintronics devices, the

research topic in the current study is important and timely for spintronics applications. However, the discussion based on the spin-swapping effect is not enough for publication in Nature Communications, as I pointed out below.

1) Whether between spin Hall effect and spin swapping effect in the spin transport dominate depend on the magnitude of spin-orbit coupling as shown in Figure 4d. However, the mean free path is also an important parameter for it (PRL 117, 036601 (2016)). If the authors will discuss the experimental results based on the spin-swapping effect, they should show the relationship between the mean free path and the torque ratio in these materials (Mn₃Sn and nonmagnetic metals). Temperature and thickness dependence should relate to the mean free path. The author should show clear evidence of spin swapping from this fundamental experiment before the discussion of the experimental results on spin torque based on spin swapping.

2) If the out-of-plane spin polarization is not caused by chiral spin structure, spin torque due to out-of-plane polarization should be observed more clearly in polycrystalline Mn₃Sn, because the in-plane polarization is related to chiral spin structure, the in-plane component should be suppressed in polycrystalline film.

3) They measured the initial large magnetic field direction dependence of angle dependence of spin torques. The torque due to in-plane and out-of-plane polarization are unchanged. Consequently, the authors claimed that the external magnetic field does not provide insight into the mechanism of p_y and p_z . However, the magnetization of Mn₃Sn directly in contact with ferromagnetic metal (Py) should be modulated by the direction of the magnetization of Py by the exchange coupling. Thus, even in a small external magnetic field for controlling the magnetization of Py, the magnetization of Mn₃Sn should be affected. It means that even in the case of the opposite large external magnetic field, the Mn₃Sn might be not fixed to the initial external magnetic field direction. Authors should consider the influence of the analysis of angular dependence of spin torques.

4) The temperature and thickness-independent behavior is completely different with similar non-collinear antiferromagnet Mn₃GaN (Nat. Commun. 11, 4671 (2020)). What is the main reason for the difference? What is the critical parameter to the emergence of the interfacial effect you observed?

In this work, B. K. Hazra et al reported interface-induced out-of-plane spin polarization (p_z) in noncollinear antiferromagnet Mn_3Sn and nonmagnets Cu, Ru, Re and Pt. By performing temperature dependent spin-torque ferromagnetic resonance (ST-FMR) measurements, this work demonstrated that the interfaces of Mn_3Sn/Py or NM/Py is the origin of p_z , and the generation interface can be ascribed to the spin swapping effect. The generation of p_z is highly desired and the physical mechanism is still not cleared up to now. This work is timely because it attempts to investigate the interfacial origin of p_z , and the results are very interesting. However, several questions need to be addressed before I formally recommend this work to be published in Nature Communications.

We thank the Referee for finding our manuscript “*timely*” and “*very interesting*”. Below we provide the answers to all the queries. Following the Referee's recommendation, we have now conducted additional experiments to strengthen aspects of the manuscript's arguments and analyses. Please see our point-by-point responses below.

1. As shown in Figure 3, p_z exists at both 0° and 90° devices. Whether p_z also exists in a device with specific angle (such as 45°)? What's the relationship between p_y p_z and the crystal orientation (i.e. device angle)?

Following the referee's remark, we have now carried out these new experiments. We find that p_y and p_z remain finite for the device with the angle $\varphi = 45^\circ$. Note that $\varphi = 0^\circ$ represents the device where I_{RF} is along the in-plane crystallographic direction $[01\bar{1}0]$ of Mn_3Sn . Our results also show that the torque due to p_y is independent of the crystal orientation/in-plane device angle whereas torques due to p_z show a small angular dependence. Angular variations of torques due to p_y and p_z are now added in the supplementary information (Fig. S10) and displayed here.

Fig. S10: Variation of (a) $\tau'_{y,AD}$, (b) $\tau'_{z,FL}$ and (c) $\tau'_{z,AD}$ as a function φ .

2. Because p_z is different for the 0° and 90° devices (as shown in Fig. 3), whether the result in Fig.4 is from 0° or 90° device should be described in the figure caption.

Thank you for pointing this out. We have added in the figure caption of Fig. 4 that the results of p_y and p_z are shown for the 0° device.

3. In this work, the ST-FMR device is the GS-type instead of the standard GSG-type (as shown in Ref. 10). Note that GS-type device may produce misleading signals or even artifacts [JMMM 505 166727 (2020)]. I strongly recommend authors to use the standard GSG-type devices to test the ST-FMR results.

We appreciate the referee's recommendation which we have followed. We have fabricated new devices and measured all these devices with the GSG measurement configuration as the referee proposed. These studies show that ST-FMR analysis using these new GSG devices does not alter the primary conclusions drawn in our manuscript. Specifically, we have confirmed that (i) the antiferromagnetic domain structure of Mn_3Sn generates a temperature-dependent in-plane anti-damping torque ($\tau_{y,AD}$) and (ii) there is a temperature-independent in-plane field-like torque ($\tau_{z,FL}$) that is not affected by the under-layer, resulting purely from the spin swapping effect.

However, we observed both in-plane field-like ($\tau_{z,FL}$) torque and an out-of-plane anti-damping-like ($\tau_{z,AD}$) torque in these new GSG device whereas only a significant $\tau_{z,FL}$ dominated in our earlier GS devices. Based on these new findings, we have revised the main text and supplementary text to reflect these new findings. A detailed comparison of the findings from GS and GSG-type devices is shown in the supplemental section (Fig. S17).

4. The discussion about the generation mechanism of p_z is not vivid enough. Thus, I recommend authors to add a schematic of the spin swapping mechanism at the interface of Mn_3Sn and Py, which is helpful for readers to follow.

We thank the referee for helping us to improve our manuscript. Following the referee's suggestion, we have modified and more clearly described the schematic in the lower panel of Fig. 1e in the main text (see below) which portrays the spin swapping process. Additionally, following the suggestion from both referees A and B, we have added a discussion section (highlighted in yellow in the main text) that specifically examines current experimental findings in relation to several theoretical proposals and provides a rationale for why the spin swapping mechanism is the most closely aligned with our experimental results.

Fig. 1e (lower panel): Schematic of spin swapping mechanism.

5. This work uses $\tau_{y,AD}/\tau_{y,FL}$ and $\tau_{z,FL}/\tau_{y,FL}$ to represent the intensity of p_y and p_z . While I recommend authors to calculate the values of spin torque efficiency of p_y and p_z , as done in Ref. 15.

We thank the referee for this suggestion. Following ref. 15 [You et. al., *Nat. Commun.* 12, 6524 (2021)], we have calculated the spin-torque efficiency of p_y and p_z which is added in the supplementary information (Section V, Fig. S8(g-h)). This is highlighted in yellow. The spin-torque efficiency due to p_y (θ_y) and p_z (θ_z) are 0.015 and 0.002 for a 12 nm thick Mn_3Sn film, respectively.

6. According to the temperature dependent experiments, the generation of py in Mn₃Sn is mainly caused by the SOC-independent AFM structure, rather than conventional SOC-dependent SHE. Thus, I think symbols of Mn₃Sn is not suitable in Fig. 4d considering the x-label is SOC.

The x-label SOC has been modified to 'SOT layer' according to the referee's recommendation.

Reviewer #2 (Remarks to the Author):

The manuscript is devoted to spin currents in epitaxial thin films of the chiral antiferromagnet Mn₃Sn interfaced with a thin layer of permalloy. By performing spin-torque ferromagnetic resonance (FMR) measurements, the authors find two types of currents – polarized in-plane and polarized out-of-plane. Based on the temperature-dependent FMR results for Mn₃Sn/permalloy bilayers and the results of studies with several other non-magnetic metals at the interface, the authors conclude that the in-plane polarized spin current is generated from the interior of the Mn₃Sn layer, while the out-of-plane polarized spin current originates from scattering at the Mn₃Sn/permalloy interface. The authors put forward the spin swapping effect as the origin of the observed out-of-plane polarized spin current.

Generation and understanding the nature of spin polarized spin currents in magnetic thin-film systems is important due to these currents being able to produce spin torques and switch magnetization of a ferromagnetic layer. Several recent publications have emphasized the importance of the out-of-plane polarized spin current, due to its efficiency in switching perpendicular magnetized films in the absence of an external magnetic field. In case of chiral antiferromagnets, such as Mn₃Sn, the origin of the out-of-plane polarized spin current has been attributed to the non-collinear antiferromagnetic order of these antiferromagnets. The authors of the manuscript challenge this interpretation, by showing that the out-of-plane polarized spin current is correlated not with the antiferromagnetic order in Mn₃Sn but rather with the strength of spin-orbit coupling at the interface. This finding is important for elucidating the origin of the out-of-plane polarized spin current capable of deterministic switching a perpendicular magnetization. I would recommend publication of this manuscript after the authors have addressed the two following issues.

We thank the referee for his/her evaluation of our manuscript and for his/her recommendation for publication. We are very pleased that the referee finds our manuscript “important for elucidating the origin of the out-of-plane polarized spin current”. Below we provide responses to the two issues the referee highlights. Following the Referee’s suggestion, we have now improved the discussion in the manuscript. Please see the details in our point-by-point replies below.

1. The authors explain their observation in terms of the spin swapping effect and even include this term in the title of the manuscript. While this explanation may be plausible, in my view, there is not enough evidence provided in the manuscript to make this interpretation solid. Other possible origins of the out-of-plane polarized spin current not related to the antiferromagnetic order may play a role, such as an interface-generated spin current, a spin-orbit self-torque due to a composition gradient in the ferromagnetic layer (e.g., due to intermixing at the interface),

an orbital Hall effect, etc. I think that the authors need to relax their statements and use the spin-swapping effect as a possible explanation rather than the proved finding. I also think that "spin swapping" should be removed from the title of the manuscript.

We thank the referee for these comments, which have proven valuable in enhancing the quality of our manuscript. We have now included a new paragraph that discusses other possible sources of spin currents and their respective spin-polarization directions. We discuss the limitations of these other theoretical models with respect to our findings. The spin swapping model remains the most plausible explanation for our findings.

According to theoretical predictions made by Amin et al. (Phys. Rev. Lett. 121, 136805 (2018)), an in-plane magnetized layer (FM) in close proximity to a non-magnetic (NM) layer generates an out-of-plane polarized spin current. However, this mechanism is valid for FM₁/NM/FM₂ tri-layer system, where the bottom FM₁ layer produces an in-plane spin-polarized current that is transmitted through the NM layer and generates out-of-plane spin current and successive torques on FM₂. On the other hand, in our particular case, we have observed an out-of-plane polarized current in Mn₃Sn/Py or NM/Py bilayer structures, which differs from the FM₁/NM/FM₂ structure. As a result, we conclude that the aforementioned effect is not responsible for our observation.

The self-torque resulting from the composition gradient or spatial distribution can't be considered significant in the 5 nm thick permalloy layer (Ni₈₀Fe₂₀) which is used in our devices. Recent observations have shown that self-torque exists in a single layer of permalloy layer below 3 nm in thickness. However, the same study also indicated that this torque was not detected in a 5 nanometer-thick Py layer. In addition, the torque in a single layer of Py shows an in-plane (p_y) polarized spin current only. [Seki et al., Phys. Rev. B 104, 094430 (2021)].

So far, all the experimental and theoretical developments of the orbital Hall effect (OHE) show only the existence of a p_y polarization which strongly depends on the magnetization of the ferromagnets. If OHE was the mechanism in our devices then p_z should strongly depend on the magnetization of the ferromagnet. Thus, we have made similar devices of Cu/Py where the Py layer was replaced by Fe or Ni. However, we see a relatively weak magnetization dependence of p_z (reduced by half from Cu/Fe to Cu/Ni) although we do find that p_y strongly depends on the magnetization (reduced 4 times from Cu/Fe to Cu/Ni) (see Fig. S18 and below). This rules out the OHE as the origin of p_z and supports the spin swapping origin of p_z . Note that in spin swapping mechanism, p_z weakly depends on the magnetization of the ferromagnets [Park, H.-J. et al. Phys. Rev. Lett. 129, 037202 (2022)].

Fig. S18: (a) $\tau'_{y,AD}$, (b) $\tau'_{z,FL}$ and (c) $\tau'_{z,AD}$ for different ferromagnets with different magnetization in Cu/Fe and Cu/Ni bilayers.

A summary of these mechanisms is given below:

Mechanism and system	Polarization	Ref.
Interface: Spin-orbit precession FM ₁ /NM/FM ₂ tri-layer	p_z	Phys. Rev. Lett. 121 , 136805 (2018)
Interface: Spin swapping FM/NM bilayer	p_z	Phys. Rev. Lett. 103 , 186601 (2009)
Interface: Spin-orbit filtering FM ₁ /NM/FM ₂ tri-layer	p_y	Phys. Rev. Lett. 121 , 136805 (2018)

Please note the findings below clearly indicate that the source of the out-of-plane spin polarization (p_z) that we observe is likely due to spin swapping:

- It was predicted that disorder has a significant impact on the magnitude of the spin swapping effect. We conducted additional experiments on Si/Cu/Py and Si/Pt/Py and compared them to Al₂O₃/Cu/Py and Al₂O₃/Pt/Py. Our findings show a relationship between the spin Hall effect and spin swapping with the disorder that is consistent with theoretical predictions shown in Fig. 3, PRL **117**, 036601 (2016) (Fig. 4 (e,f) in main text and below).
- It has been predicted theoretically by Park et. al. (Phys. Rev. Lett. **129**, 037202 (2022)) that p_z generated by spin swapping shows a weak magnetization dependence which we have found in our studies (Fig. S18).
- The existence of the out-of-plane polarization in other NM/FM bilayers (e.g. Cu, Ru, Re, Pt/Py) shows that the origin is extrinsic in nature.

Fig. 4(e,f): $\tau_{y,AD}/\tau_{z,AD}$ and $\tau_{y,AD}/\tau_{z,FL}$ for Cu/Py and Pt/Py grown on Si/SiO₂ and Al₂O₃ substrates. The interplay of spin swapping and spin Hall effect as a function of the disorder can be seen clearly.

2. Previous experimental results have shown that the spin-orbit torque generated in a Mn₃Sn/ferromagnet bilayer can be reversed with the reversal of the antiferromagnetic order, i.e. it is odd with respect to time reversal symmetry. The authors are saying however that their torques are even under magnetic field. Why the results presented in the manuscript are different from the previous observations?

We appreciate this question. It is worth noting that the previous report on single crystal bulk Mn₃Sn/Py bilayer shows that p_z is odd under a magnetic field. Based on this dependence, it was claimed that the origin is related to antiferromagnetic structure. However, temperature-dependent evolution of the torque due p_y and p_z was not reported in those studies. Our study, on the other hand, demonstrates that p_z is independent of temperature which clearly rules out an antiferromagnetic origin. We find that the p_z is caused by spin swapping, which is consistent with an even magnetic field dependence.

We also believe that the recent ST-FMR-based analysis on the single crystal bulk Mn₃Sn/Py bilayer [Kondou et al Nat. Commun. **12**, 6491 (2021)] is a little puzzling. In ST-FMR analysis, $V_s \propto \frac{dR}{d\varphi} \cos(\varphi)$ and $V_A \propto \frac{dR}{d\varphi} \cos(\varphi)$, where φ is the angle between the external magnetic field and current direction/x axis [Liu et. al., Phys. Rev. Lett. 106, 036601 (2011)]. Surprisingly, Kondou et. al. considered, $V_s \propto \frac{dR}{d\varphi} \sin(\varphi)$ and $V_A \propto \frac{dR}{d\varphi} \sin(\varphi)$ where φ is the angle between the external magnetic field and the y-axis whereas the current remains along the x-axis. Why they have taken a different φ compared to the conventional coordinate system is not clear from their study. However, if you choose a different origin then this can give a different result.

Reviewer #3 (Remarks to the Author):

B. K. Hazra et al., reported the experimental study of out-of-plane polarized spin current by spin swapping. The spin torque due to in-plane and out-of-plane spin polarization was measured utilizing the ST-FMR method in Py/Epitaxial thin Mn₃Sn film bilayer. Temperature dependence above Neel temperature suggests the in-plane spin polarization is related to the chiral spin structure, however, the out-of-plane polarization is not related to the spin structure. By control experiments using several nonmagnetic metals, they conclude that the out-of-plane components are caused by the spin-swapping effect in Mn₃Sn. Because the noncollinear antiferromagnet could enable new functionality in spintronics devices, the research topic in the current study is important and timely for spintronics applications. However, the discussion based on the spin-swapping effect is not enough for publication in Nature Communications, as I pointed out below.

We appreciate the referee's assessment of our work as "important and timely for spintronics applications." We have provided point-by-point responses to all of the referee's questions/comments below. We have now expanded our discussion to include a comparison of several theories to explain our observations.

1) Whether between spin Hall effect and spin swapping effect in the spin transport dominate depend on the magnitude of spin-orbit coupling as shown in Figure 4d. However, the mean free path is also an important parameter for it (PRL 117, 036601 (2016)). If the authors will discuss the experimental results based on the spin-swapping effect, they should show the

relationship between the mean free path and the torque ratio in these materials (Mn₃Sn and nonmagnetic metals). Temperature and thickness dependence should relate to the mean free path. The author should show clear evidence of spin swapping from this fundamental experiment before the discussion of the experimental results on spin torque based on spin swapping.

We thank the referee for these interesting suggestions. According to the above-mentioned paper (Phys. Rev. Lett. **117**, 036601 (2016)), the spin Hall effect dominates in the diffusive limit ($d \gg \lambda$), but in the Knudsen regime ($d \lesssim \lambda$) spin swapping dominates. Therefore, the mean free path is important in controlling the relative magnitude of these effects. The same paper also demonstrates that spin Hall and spin swapping will coexist in a system and the relative strength will vary depending upon the degree of disorder and the magnitude of the spin-orbit coupling. We indeed find that the magnitude of the torques due to out-of-plane polarization (p_z) depend strongly on the magnitude of the disorder.

We have performed additional experiments with GSG-type devices on Si/Cu/Py, Si/Pt/Py and compared them with Al₂O₃/Cu/Py, Al₂O₃/Pt/Py to show the interplay between spin Hall and spin swapping effect as a function of disorder as predicted theoretically in fig. 3, Phys. Rev. Lett. **117**, 036601 (2016). When the disorder is greater [Si/Cu/Py and Si/Pt/Py] the spin-Hall effect dominates and the spin swapping effect dominates when the disorder is less [Al₂O₃/Cu/Py, Al₂O₃/Pt/Py] (See Fig. 4(e,f)). These results and results on other non-magnetic/Py bilayers give more evidence of the interplay between spin swapping and the spin Hall effect. We have now discussed this in detail in the main text.

Please note that since the spin diffusion length in Mn₃Sn is reported to be less than 1 nm [Muduli et al., Phys. Rev. B 99, 184425 (2019)], it's difficult to perform any experiments below 1 nm of Mn₃Sn. Therefore, a thickness-dependent study in our case will not be able to see any significant difference as a function of thickness.

Fig. 4(e,f): $\tau_{y,AD}/\tau_{z,AD}$ and $\tau_{y,AD}/\tau_{z,FL}$ for Cu/Py and Pt/Py grown on Si/SiO₂ and Al₂O₃ substrates. The interplay of spin swapping and spin Hall effect as a function of the disorder can be seen clearly.

2) If the out-of-plane spin polarization is not caused by chiral spin structure, spin torque due to out-of-plane polarization should be observed more clearly in polycrystalline Mn₃Sn, because

the in-plane polarization is related to chiral spin structure, the in-plane component should be suppressed in polycrystalline film.

Following the referee's suggestion, we have now grown polycrystalline Mn_3Sn on Al_2O_3 substrate without a Ru buffer layer. As the referee has pointed out, the anti-damping ($\tau_{y,AD}$) torque due to the in-plane polarization (p_y) is not present and V_S shows a very unusual angular variation (Fig. S15). It is worth noting here that the epitaxial Mn_3Sn (0002) film does not show an anomalous Hall effect (AHE) whereas polycrystalline Mn_3Sn exhibits a large AHE effect. The AHE might contribute to SHE which makes the analysis more complicated.

On the other hand, the angular dependence of V_A indicates the presence of a $\tau_{z,AD}$ ($\tau'_{z,AD} = 0.028$) due to p_z and Oersted field dominating field-like torque. We believe that there are two reasons which might affect the p_z . Firstly, from structural analysis, we find that the magnetic properties of Py films grown on polycrystalline Mn_3Sn are different, e.g higher coercivity. Secondly, we find that the interface roughness increases significantly for growth on polycrystalline Mn_3Sn thin films as compared to epitaxial Mn_3Sn thin films.

Fig.S15: Angular variation of V_S and V_A for the polycrystalline $\text{Mn}_3\text{Sn}/\text{Py}$ bilayer.

3) They measured the initial large magnetic field direction dependence of angle dependence of spin torques. The torque due to in-plane and out-of-plane polarization are unchanged. Consequently, the authors claimed that the external magnetic field does not provide insight into the mechanism of p_y and p_z . However, the magnetization of Mn_3Sn directly in contact with ferromagnetic metal (Py) should be modulated by the direction of the magnetization of Py by the exchange coupling. Thus, even in a small external magnetic field for controlling the magnetization of Py, the magnetization of Mn_3Sn should be affected. It means that even in the case of the opposite large external magnetic field, the Mn_3Sn might be not fixed to the initial external magnetic field direction. Authors should consider the influence of the analysis of angular dependence of spin torques.

We thank the referee for these comments. Please note that ST-FMR measurements are carried out below 1000 Oe in which only the magnetization of Py is changed. From the magnetization vs. fields measurement on Mn_3Sn , one can see that the in-plane magnetization switches above 5000 Oe [See supplementary, Fig. S5]. So, during the ST-FMR measurements, the magnetization of Mn_3Sn remains unchanged, hence the analysis should be similar to the Pt/Py

case. When we apply a large magnetic field of +/- 7T, both Mn₃Sn and Py magnetizations will be changed but during ST-FMR measurements only Py magnetization is manipulated.

4) The temperature and thickness-independent behavior is completely different with similar non-collinear antiferromagnet Mn₃GaN (Nat. Commun. 11, 4671 (2020)). What is the main reason for the difference? What is the critical parameter to the emergence of the interfacial effect you observed?

We thank the referee for this comment. First, we would like to point out that the crystal structures of Mn₃GaN and Mn₃Sn are very different: one is cubic and the other is hexagonal. In the cubic, there are 8 distinct (111) planes and corresponding antiferromagnetic domains that are oriented at different angles to each other whereas in the hexagonal Mn₃Sn case there are 6 distinct domains all with the Mn moments parallel to a single (0001) plane. In fact, the magnetic properties of the cubic and hexagonal Mn₃Sn are different. Thus, it is difficult to compare these systems directly.

Moreover, the data presented in the Mn₃GaN paper is not self-consistent in the entire temperature range. The neutron diffraction and magnetization measurements show that net moment due to a tetragonal distortion decreases with an increase in the measurement temperature and magnetization drops to zero/nearly zero at T_N [supplementary figures 3 and 5, Nat. Commun. 11, 4671 (2020)]. The claim of the antiferromagnetic origin of p_x and p_z and the conventional origin of p_y is demonstrated from the temperature dependence in the range of 300-380 K (Fig 4 in the main text, Nat. Commun. 11, 4671 (2020)). The temperature dependence of the torques due to p_x and p_z in the temperature range 25-300 K shows an opposite behavior to the magnetization temperature dependence [supplementary figures S8 (a-c) (Nat. Commun. 11, 4671 (2020))]. Moreover, the torque due to p_z approaches zero at 25 K whereas the magnetization remains finite. These results clearly demonstrate that although torques due to p_x and p_z follow the magnetization temperature dependence at high-temperature (300-380 K), it does not follow at low temperatures which complicates the understanding of unconventional torque related to antiferromagnetism.

In addition, Mn₃GaN [K. Takenaka et. al. Appl. Phys. Lett. 87, 261902 (2005)] shows a large negative thermal expansion and a large change in lattice constant across the magnetic ordering temperature whereas such change in lattice constant was not observed in Mn₃Sn (Y. Song et. al. Chem. Mater. 2018, 30, 6236–6241]. This might also influence the temperature dependence of conventional and unconventional torques. More temperature-dependence spin-orbit torque experiments on other antiferromagnetic materials might be helpful to make a generalized conclusion.

The critical parameter to observe the interface effect could be the crystallinity and interface of Mn₃Sn/Py and non-magnetic/Py bilayers. When the system or interface is more disordered, spin swapping becomes small and then other effects such as p_z due to magnetic spin Hall effect can be observed. This could be the case of single crystal bulk Mn₃Sn which is interfaced with a thin film of Py [K. Kondou et. al. Nat. Commun. 12, 6491 (2021)] where the interface will be poor compared to the thin film of our Mn₃Sn/Py bilayers.

Reviewers' Comments:

Reviewer #1:

Remarks to the Author:

The authors addressed my previous comments and questions very well. I also found that besides my basically positive evaluation, there is one positive and one negative reports. I also read the response to the negative one. It seems that the authors have addressed the questions reasonably. Thus I recommend the publication of this work as it is.

Reviewer #2:

Remarks to the Author:

The authors have adequately responded to the comments of the referees, and I recommend publication of the manuscript as is.

Reviewer #3:

None